# Treatment Realities of Headache Disorders in Rural Germany by the Example of the Region of Western Pomerania

**DOI:** 10.3390/brainsci11070839

**Published:** 2021-06-24

**Authors:** Anne Thiele, Sebastian Strauß, Anselm Angermaier, Lara Klehr, Luise Bartsch, Martin Kronenbuerger, Sein Schmidt, Robert Fleischmann

**Affiliations:** 1Department of Neurology, University Medicine Greifswald, 17475 Greifswald, Germany; anne.thiele@stud.uni-greifswald.de (A.T.); sebastian.strauss@uni-greifswald.de (S.S.); anselm.angermaier@uni-greifswald.de (A.A.); lara.klehr@stud.uni-greifswald.de (L.K.); luise.bartsch@med.uni-greifswald.de (L.B.); martinkro2@hotmail.com (M.K.); 2Medical School OWL, University of Bielefeld, 33615 Bielefeld, Germany; 3Clinical Research Unit, Charité Campus Mitte, Berlin Institute of Health, 10117 Berlin, Germany; sein.schmidt@charite.de

**Keywords:** migraine, headache, disability, treatment, health care delivery, health care quality, outpatient, quality of life

## Abstract

(1) Background: Headache disorders are among the most disabling medical conditions but the supply with experienced providers is outpaced by the demand for service. It is unclear to what extent particularly patients in rural regions are affected by limited access to comprehensive care. Furthermore, it is unknown what role general practitioners (GPs) play in headache care. (2) Methods: First-time consultations to a specialised headache clinic at a tertiary care centre were asked to participate. Their socio-demographic background, general and headache-specific medical history, disability and quality of life (QoL) were assessed. Additionally, 176 GPs in neighbouring districts were contacted regarding headache management. (3) Results: We assessed 162 patients with first-time consultations (age 46.1 ± 17.0 years, 78.1% female), who suffered from migraine (72%), tension type, cluster and secondary headaches (each 5–10%). About 50% of patients received a new headache-diagnosis and 60% had treatment inconsistent with national guidelines. QoL was significantly worse in all domains compared to the general population. About 75% of GPs see headache patients at least several times per week, and mostly treat them by themself. (4) Conclusions: More than every second headache patient was neither correctly diagnosed nor received guideline adherent treatment. Headache-related disability is inferior to what is expected from previous studies. Access to specialised health care is more limited in rural than in urban regions in Germany and GPs request more training.

## 1. Introduction

Migraine and other headache disorders range among the most prevalent and most disabling diseases worldwide [1]. Suffering from a headache disorder significantly impairs multiple domains of personal life such as employment, physical, social and family activities [2] and is associated with substantial costs for health care providers and society [1]. Despite the individual and economic impact of headache disorders, the World Health Organization recognises that “they are under-recognized, under-diagnosed and under-treated” [3]. The unmet medical need was recently confirmed in a German population of migraineurs who utilised a specialised headache clinic in a metropolitan area [4]. About 36% of patients were not treated according to national guidelines and 53% never received a preventive treatment despite clear indication [4]. This finding is particularly surprising given the density of the medical infrastructure in the geographic region assessed and since 90% of the patients previously consulted a general practitioner (GP) and 75% of patients consulted a neurologist [4]. In line with this notion, the Vancouver Declaration on Global Headache Patient Advocacy found that lifting the burden of headache not only requires effective therapies but that patients should have reliable access to comprehensive medical care [5]. Ineffective delivery of specialised care might be explained by insufficient training of healthcare providers, but also lack of awareness for the incapacitating character of severe headache disorders such as migraine [6]. Insufficient interaction and exchange between general and specialised health care providers may pose another limitation, but this is less well studied. GPs are on the frontlines of primary care and often first approached by headache patients, thus understanding their role in headache care may be a critical step to address missed opportunities. Management of headache disorders in rural areas may be particularly affected by management in primary care since access to specialists is limited.

In order to fill this gap, this study aims to clarify treatment realities of headache disorders in rural Germany and their association with primary care in two steps. We first hypothesise that the lower availability of specialist care in rural regions negatively impacts on the adherence of headache management with national guidelines. This hypothesis is tested by investigating data from a headache population treated in a specialised headache clinic in Western Pomerania, one of the most rural areas of Germany [7], and comparing these results to treatment patterns observed in an urban area [4]. We then conducted an exploratory investigation of routine headache management among regional GPs and discuss their possible contribution to observed treatment patterns.

## 2. Materials and Methods

The study was approved by the local ethics committee (protocol number BB 161/18) and conducted in line with the Declaration of Helsinki in its latest revision. Data were acquired in a specialised headache clinic, which is affiliated with the Department of Neurology at the University Medicine of Greifswald. The headache clinic provides an outpatient service to headache patients referred for consultation by their primary care physician or medical specialist. Data collection and evaluation required written informed consent from either patients or GPs.

### 2.1. Part 1—Headache and Sociodemographic Characteristics of First-Time Consultations to the Headache Clinic

All first-time consultations to the headache clinic between August 2018 and December 2019 were considered without exclusion criteria to avoid a selection bias. All patients routinely filled-in a questionnaire as part of their initial evaluation, which allowed for patient self-report of general information about age, gender, social environment, their living situation, family status and profession and their medical history. Additionally, detailed information about their headache disorder was gathered, including duration of the headaches in years, headache days per month, headache duration per attack, family history, previous contacts to specialists and information about the use of acute and prophylactic medication. Moreover, all patients were assessed regarding their functional abilities and quality of life (QoL) utilising the Migraine Disability Assessment (MIDAS) and Patient-Reported Outcomes Measurement Information System Profile 29 (PROMIS-29), which assesses seven QoL domains as well as general QoL [8]. Data obtained provide the treating physician with a bio-psycho-social context of the patient’s living situation and conditions, and the headache disorder. First-time consultations usually take about 45–60 min to provide the patient with a solid diagnosis, or suggestions for further diagnostic work-up, and a treatment plan. Furthermore, as part of this study, the headache specialist in charge documented whether or not there was an indication for preventative treatment according to national guidelines and whether or not the indication was met.

### 2.2. Part 2—Assessment of Routine Headache Management by Primary Health Care Providers

We additionally contacted 176 GPs in the three neighbouring districts of Western Pomerania (Vorpommern-Rügen, Vorpommern-Greifswald, Mecklenburgische Seenplatte) including a consent form and study information. These general practitioners work independent of the headache clinic but constitute the majority of primary care physicians that can directly refer their patients for consultation. Their number vastly outpaces that of neurologist or pain specialists in the vicinity. The aim was to further explore the role of GPs as the first contact for patients with headache disorders. We provided a questionnaire with seven multiple-choice questions assessing their routine management of headache patients, including questions about the estimated frequency of patients they treated for headache disorders, how they dealt with particular primary headaches, knowledge of and reasons to consult specialised healthcare providers and their interest in further education. An English translation of the questionnaire can be found as Appendix A. GPs returned questionnaires anonymously, which was accomplished to ascertain data confidentiality.

### 2.3. Data Evaluation and Statistics

All patient data were pseudonymised and entered into an electronic data capture system. Data provided by GPs were anonymised and digitalised in analogy to patient data. All descriptive analyses and statistical evaluations were accomplished using Statistical Package for the Social Sciences (SPSS v25.0, IBM, Armonk, NY, USA).

### 2.4. Evaluation of Data from First-Time Consultations to the Headache Clinic

Patient data were first investigated using descriptive statistics to assess sociodemographic factors (age, gender, living and working situation), substance use (smoking, alcohol, other drugs), headache characteristics (headache history, frequency, MIDAS) and quality of life (PROMIS-29) of patients presenting for the first time to the outpatient clinic. Results from PROMIS-29 evaluations were transformed into population standardised T-scores using the database provided by the German PROMIS national reference center. T-Scores of 50 represent by definition the reference population mean, standard deviations are standardised to scores of 10, i.e., two standard deviations above mean would yield a T-score of 70.

### 2.5. Statistics of Data from First-Time Consultations to the Headache Clinic

Continuous data were analysed for normal distribution using histogram plots before performing descriptive and inferential statistics. Unless stated differently, normal distribution was confirmed. Sociodemographic factors and substance use were analysed regarding their influence on headache frequency and headache duration using a generalised linear regression model of main effects including nominal and ordinal data as factors and continuous data as covariates. Predictors are presented with their beta coefficient and 95% confidence intervals (95%CI) in square brackets. Medians and interquartile ranges (IQR) of the seven PROMIS domains (i.e., anxiety, depression, fatigue, sleep, physical and social functioning, pain interference) and global QoL score were calculated given that histograms revealed a non-normal distribution of data. One-sample Wilcoxon signed rank tests were used to evaluate whether the headache population’s results differ from the nearest T-Score and standardised standard deviation. Finally, we evaluated whether PROMIS results correlated with headache characteristics, and which PROMIS domains correlate most with the MIDAS as a widely used tool to assess disability of migraineurs, using linear analyses and the Pearson correlation coefficient.

### 2.6. Evaluation and Statistics of Survey Responses from General Practitioners

Descriptive analyses were performed using contingency tables since multiple choice data generally returns response frequencies. Statistical significance of differences of responses were either evaluated using chi-square tests (nominal data, binary responses) or univariate analysis of variance (ANOVA, continuous data).

## 3. Results

### 3.1. Part 1—Demographic and Headache Characteristics of First-Time Outpatient Consultations

There were 162 first-time consultations (age 46.1 ± 17.0 years, 78.1% female) in the study period. Detailed patient characteristics are summarised in Table 1.

Referrals to the headache clinic were carried out by GPs (78.8%), neurologists (17.9%), psychiatrists (1.3%), dentists (1.3%) and gynaecologists (0.7%). The prevalence of primary headaches was 71.8% migraine (22% of these chronic migraine), 9.2% tension-type headache, 6.1% cluster headache, 8.0% other primary headache syndromes and 4.8% secondary head or facial pain syndromes. Seventeen percent of patients presented with a medication-overuse headache. Headache frequency was on average 16.8 ± 8.2 days per month. Patients suffered for 17.1 ± 13.8 years from the headache syndrome leading to the consultation, yet 46% of patients were given a first-ever or new headache diagnosis as a result of the consultation. About 60% of the patients did not receive acute or prophylactic treatment according to national guidelines. 82% of patients met the indication for a prophylactic treatment, but only 52% of these received one. Patients with the correct diagnosis before consultation had significantly higher odds to receive guideline adherent treatment (OR 9.2 [95%CI: 3.7–22.4]).

### 3.2. Part 1—Influence of Sociodemographic Factors on Headache Frequency

Headache frequency per month was significantly influenced by the living situation. Living with the family was associated with −7.2 [95%CI: −1.1–−13.5] (*p* = 0.022) headache days per month as compared to living alone. Age, gender, family status, number of children and profession did not influence headache days per month. Being a non-smoker was associated with −3.9 [95%CI: −0.2–−7.7] (*p* = 0.042) headache days per month while alcohol consumption or drug use were no significant predictors.

### 3.3. Part 1—Disability and Quality of Life in First-Time Outpatient Consultations

The distribution of the MIDAS score was non-normal. Its median value was 42.5 days (IQR 22.5–75.3, range 0–180). The results of the PROMIS evaluations are summarised in Table 2. In brief, patients performed significantly worse in all domains compared to a German reference population, i.e., their T-scores for anxiety, depression, fatigue, sleep disturbance and pain interference were significantly higher and those for physical and social functioning, and global health, significantly lower. More severe headache frequency was negatively correlated with physical (ρ = −0.25, *p* = 0.005) and social functioning (ρ = −0.24, *p* = 0.005), and positively correlated with pain interference (ρ = 0.25, *p* = 0.004). The MIDAS score significantly correlated with all PROMIS domains except for fatigue and sleep disturbance, however correlations with anxiety and depression T-scores were only weak (i.e., ρ < 0.30).

### 3.4. Part 2—Treatment Patterns in Primary Care

About 45% of GPs (*n* = 76) contacted participated in the survey. 72% of GPs reported to treat headache patients either daily (25%) or several times a week (47%). Treatment patterns for individual headache disorders are summarised in Figure 1. In brief, GPs always or often treat migraine and tension type headache by themself, while patients with symptomatic headache syndromes or unknown diagnoses are more likely to be referred to a specialist. Reasons for referral to a specialist other than the headache diagnosis were insufficient treatment response (84%) and patient’s request (63%). Lack of therapeutic options (31%) and inability to provide treatment according to guidelines (12%) rarely caused referrals. GPs reported that they would most likely refer headache patients to either a general neurologist (95%) or pain specialist (48%). Only 23% of GPs would refer a patient to a headache specialist, however 61% of GPs reported to know neither a headache specialist nor a specialised headache clinic in their region. About 85% of GPs would appreciate further training for the treatment and diagnosis of headache disorders.

## 4. Discussion

This study conducted in patients presenting to a specialised headache clinic not only replicates evidence for the under-recognition and under-diagnosis of primary headache disorders, but our findings also extend on previous reports by providing insights into predictors of headache frequency and showing that all domains of bio-psycho-social health are affected by headache frequency. Finally, observed treatment patterns in primary care provide possible explanations for inadequate headache care and starting points for alleviating patient burden.

### 4.1. Comparison to Headache Treatment in Urban Areas of Germany and Internationally

Under-treatment and under-diagnosis of primary headache disorders are well documented phenomena. The range of correct diagnoses in patients ranges internationally from 27% in an Italian multicentre population to 56% in a US population [9,10]. We were unable to find similar data for Germany and thus provide first evidence that the proportion of patients with an adequate headache diagnosis is in the upper international range but equally insufficient. This is even more surprising given that the mean duration patients suffered from their headaches was more than 17 years. The diagnostic delay was accompanied by inadequate acute and prophylactic treatment since patients with a correct diagnosis were 9-times more likely to receive guideline adherent treatment. About 50% of the patients with an indication for prophylactic treatment never received any before consultation in our department. Ziegeler et al. found that even 61% of patients of their urban headache population did not receive a prophylactic treatment despite indication according to national guidelines [4]. This apparent superiority of guideline adherence in our study does not hold true when the total patient population is considered, which renders the proportion of patients without guideline-adherent preventative use in our population even higher, i.e., 39% vs. 34% (see Table 3 for comparison of the studies). The higher rate of medication overuse headaches (17% vs. 9%) furthermore indicates inadequate use of acute medication, which is supported by lack of triptans for acute treatment of migraine attacks. Adding another 10% of patients with inadequate acute medication use, 60% of our patient population did not receive guideline adherent treatment. While Ziegeler et al. did not provide information on acute medication, two Italian studies found that triptan use in migraineurs is suboptimal and one-year persistence is lower than 50% [11,12]. In summary, our data support that an under-treatment of 50–60% of patients should be expected in a German headache population. This is particularly unfortunate because it was shown that adequate treatment can lead to an improvement in more than 70% of patients [4]. Sociodemographic factors did not reveal a pattern that would provide treating physicians with risk factors for the identification of severely affected patients. Yet, we were able to show that being correctly diagnosed is a significant predictor of adequate treatment, and thus screening tools may pose a suitable and viable option to enhance headache treatment. Short instruments such as the ID Migraine^TM^ provide a sensitivity of about 98% for the identification of migraine with only three questions that can be easily implemented in any outpatient setting [13].

### 4.2. Disability and Quality of Life in First-Time Outpatient Consultations

Already in 1946, the World Health Organisation defined health “as a state of complete physical, mental and social well-being and not merely the absence of disease or infirmity” [14]. In line with this notion, headache patients are not only affected by pain itself but consider their overall health, daily activities, working abilities, social and family life significantly impaired due to their headaches [15]. While headache severity accounts for some of the variance, Malone et al. found that migraineurs are additionally affected by negative life events due to psycho-emotional, e.g., worries to disappoint people, and social-interactive, e.g., impact on professional advancement, stress [16]. Two large studies conducted, the American Migraine Prevalence and Prevention (AMPP) and Chronic Migraine Epidemiology and Outcome (CaMEO) study, evaluated measures of the associated patient burden among migraineurs, which help put our results into context [17,18]. The median MIDAS scores in both studies were 3–7 in episodic and 32–45 in chronic migraineurs. The median MIDAS score in our population was about 43 with a 25% quartile of 23, which indicates that patients presenting for the first time show significantly higher disability than one would expect in a general population of migraineurs [19]. We were able to show that the MIDAS score moderately corelated with physical and social functioning, and pain interference, but that it not sufficiently represents other domains of health. The PROMIS-29 revealed that patients suffer from fatigue and sleep disturbances and are additionally affected by affective disorders. This is expected given a prevalence of depression and anxiety disorders in 40–50% of cases [20]. In summary, 75% of patients revealed a lower global health than the average population. To our knowledge, this is the first study to demonstrate this disease independent multi-domain impairment of QoL in a German population of headache patients. These results underline that patients require specialised treatment to approach all domains of headache-related disability and impaired bio-psycho-social health. Importantly, headache frequency alone did not even show a modest correlation with most of the QoL domains, which additionally highlights the need for specialist care and routine administration of instruments enabling assessment of these domains [5,21]. Specialists can provide access to multi-modal interventions that were shown to provide moderate to strong effect sizes to enhance affective symptoms, QoL and disability [22].

### 4.3. Standard of Headache Care Provided by General Practitioners in the Study Population

We were surprised by the high participation rate and the response behaviour of GPs suggests a strong motivation and interest in headache management. This may be due to high frequency that GPs encounter headache patients (that is mostly several times per week) and that treatment seems to be unsatisfactory in some cases. Indeed, another study found that one in twenty patients presenting to the GP in Germany complains about headaches [23]. In contradiction to poor guideline adherence, most GPs treat patients (in particular migraineurs) by themself and incapability of guideline adherent treatment is rarely a reason for GPs to refer a patient to a headache specialist. However, it is important to note that the Eurolight study found that treatment by GPs is superior to self-medication. Additionally, this study showed that having seen a GP for headaches is a predictor for access to specialist care. Furthermore, and as revealed by our study, GPs were highly interested in receiving training about headache management [24]. Yet, treatment patterns did not significantly differ between tension type headache patients and migraineurs in our population, hence strategies to inform GPs about the debilitating character of migraine on multiple domains and to increase their awareness could be important steps for optimising care for headache patients. Cooperative networks including general practitioners, neurologists, pain specialists and associated disciplines are a suitable platform to streamline headache management and provide health care delivery as needed [9].

### 4.4. Limitations

It remains unclear to what extent our study population represents the general population in the geographical region studied, yet we believe that this bias should be minor since no further exclusion criteria were defined and the study period comprised 12 months, which should account for fluctuations due to GP availability and waiting lists for consultations. It is furthermore possible, that the rate and interest in referrals by GPs to a specialist would have been higher if there was a longer history and knowledge of a specialised headache clinic. The outpatient clinic affiliated with the Department of Neurology was founded about two years before the study period and there was no comparable service available. Hence, treatment patterns may be different with more years having passed. If this was true, it would underline the necessity and impact of specialised headache services for the management in the surrounding area. Another possible limitation is that GPs were not interviewed directly but received only a short questionnaire in order to enhance return rates. This impedes drawing more detailed conclusions from the responses, yet it provides a surprisingly solid starting point to further investigate factors influencing headache management in rural areas and primary care.

## 5. Conclusions

Our study revealed that about 50–60% of headache patients, 75% of which being migraineurs, presenting to a specialised headache clinic are neither correctly diagnosed nor receive guideline adherent treatment, even though they suffer on average for more than 17 years from headaches. Headache-related disability was inferior to what is expected from previous studies and QoL was below population average on all domains investigated. Patients are generally treated by GPs, indicating that access to specialised health care delivery is worse in rural as compared to urban regions in Germany. GPs are generally interested in and regularly approached for management of headaches. However, GPs request more training and our data support that awareness for the debilitating impact of headache disorders on multiple domains of bio-psycho-social health is required.

## Figures and Tables

**Figure 1 brainsci-11-00839-f001:**
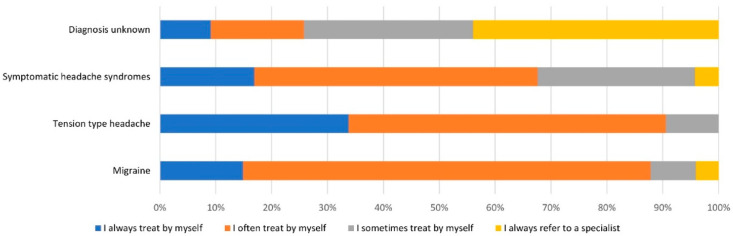
Treatment patterns reported by general practitioners. Seventy-six general practitioners (GP) from the neighbouring districts of the specialised outpatient clinic reported, which patients they usually treat by themselves and which patients they usually refer to a specialist. Migraine is in 86% of cases either always (*n =* 11) or often (*n =* 54) treated by general practitioner. Tensions type headaches are equally frequently, i.e., 88%, treated in the majority of cases by the GP. In contrast, patients with symptomatic headaches or an unknown diagnosis are more often not treated by GPs but referred in 30% or 64% of cases, respectively.

**Table 1 brainsci-11-00839-t001:** Sociodemographic and headache characteristics of first-time consultations grouped by their headache diagnosis. The vast majority of patients was diagnosed with a migraine. Migraineurs tended to be younger than patients with other primary or secondary headache disorders, however, standard deviations are large and indicate a heterogeneous patient population. The frequency of students among migraineurs was larger than in other headache disorders, which probably explains the lower rate of children. Other sociodemographic characteristics were comparable between groups. Missing values are due to missing data, i.e., patients did not wish to disclose that information to their treating physician.

	Migraine		Tension-Type Headache		Cluster Headache		Other Primary, Sedcondary Headache or Facial Pain Syndromes	
Number of patients	116		15		10		21	
Age	43.4 ± 16.3		52.0 ± 20.3		54.67 ± 14.6		53.33 ± 15.8	
Gender	86% female		64% female		33% female		67% female	
Headache days per month	14.57 ± 7.6		23.42 ± 7.7		22.67 ± 8.7		24.75 ± 8.2	
Headache duration in years	19.39 ± 13.4		4.75 ± 4.3		22.33 ± 15.8		7.36 ± 11.5	
Marital status	Single	31%	Single	36%	Single	22%	Single	16%
	Married	50%	Married	43%	Married	56%	Married	42%
	Widowed	1%	Widowed	7%	Widowed	11%	Widowed	0%
	Divorced	16%	Divorced	7%	Divorced	11%	Divorced	21%
Number of children	1.4 ± 1.2		2 ± 0.8		1.8 ± 1.5		1.9 ± 1.1	
Living situation	Alone	17%	Alone	43%	Alone	11%	Alone	23%
	Shared flat	21%	Shared flat	21%	Shared flat	33%	Shared flat	31%
	Partner	48%	Partner	29%	Partner	56%	Partner	31%
	Family	9%	Family	0%	Family	0%	Family	0%
Work status	Employed	35%	Employed	36%	Employed	33%	Employed	36%
	Retired	19%	Retired	36%	Retired	33%	Retired	32%
	Student	14%	Student	7%	Student	0%	Student	5%
	Unemployed	7%	Unemployed	0%	Unemployed	11%	Unemployed	5%
	Others	14%	Others	0%	Others	0%	Others	0%

**Table 2 brainsci-11-00839-t002:** Quality of life in patients presenting to the specialised headache outpatient clinic. Headache patients were self-administered an assessment of quality of life (QoL) through these patients reported outcome measurement information system profile with 29 items (PROMIS-29), which provides a disease independent estimate of 7 domains of and global QoL. By definition, the population mean is a T-score of 50 and standard deviations are standardised to a T-score of 10 (e.g., a T-score of 70 indicates two standard deviations above population mean). Median T-scores of the headache population were tested for statistical difference against a T-score of 50, i.e., whether headache patients score significantly worse or better than the mean population. Pain interference was also tested against a T-score of 60. We found that headache patients perform significantly worse in all PROMIS domains and that particularly pain interferes with their QoL. */** = difference statistically significant against a T-score of 50 or 60.

	Anxiety	Depression	Fatigue	Sleep Disturbance	Physical Functioning	Social Functioning	Pain Interference	Global Quality of Life
PROMIS T-Score (Median)	55.8 *	53.9 *	56.1 *	56.1 *	45.3 *	44.2 *	63.8 **	45.1 *
PROMIS T-Score (IQR: 25–75%)	51.2–61.4	49–61.8	51–62.7	48.4–61.7	37–56.9	40.5–50	58.5–66.6	40.3–50

**Table 3 brainsci-11-00839-t003:** Overview of the population characteristics investigated in the current study and its comparison to a population in an urban area. It can be seen that patients in the rural area tended to be older, more severely affected in terms of medication overuse and headache days and less likely to receive guideline adherent prophylactic treatment. In apparent contradistinction, the rate of chronic migraineurs seems to be lower. This is resolved by the fact that the current study also investigated other headache disorders, including chronic tension type headache.

	Current Study	Urban Cohort [4]
Period of data acquisition	August 2018–December 2019	2010–2018
Number of included patients	162	1935
Age	46.1 ± 17.0 years	37.3 ± 13.3
Gender (female)	78.1%	81.6%
Percentage of migraineurs with chronic migraine	22.0%	29.1%
Headache days/month	16.8 ± 8.2	12.1 ± 9.6
Rate of patients with medication-overuse headache	17.0%	9.2%
Rate of patients NOT receiving preventatives according to guidelines	39%	34%

## Data Availability

The data presented in this study are available on request from the corresponding author. The data are not publicly available due to data protection regulations.

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
