# Peer review of "Treatment Realities of Headache Disorders in Rural Germany by the Example of the Region of Western Pomerania"

_brainsci, 2021, doi:10.3390/brainsci11070839_

Round 1

Reviewer 1 Report

The study aims to evaluate headache management in a sample of patients, but the number of patients is not indicated and the general characteristics are not described in a descriptive table. The number of patients interviewed per doctor is not indicated. No table shows the comparison with the population of the urban area. 
The study should be improved with tables describing the characteristics of the patients and the comparison with the control population.

Reviewer 2 Report

Authors assessed the treatment of headaches in a real-life rural setting. Authors found that less than one half of patients with headache was correctly managed and treated in that population; besides, the treatment of headaches was mostly managed by general practitioners rather than by headache specialists.

Several points of the Methods and Discussion can be improved.

1 - Authors state in the Methods that they included “all first-time consultations in the headache clinic”. The “headache clinic” should be better defined. Besides, the relationship between the “headache clinic” and general practitioner consultations is not clear. Did the Authors mean to compare the characteristics of patients visited in the headache clinic with those of patients visited in general practice? Were there any patients visited by general practitioners who were referred to the headache clinic during the study period?

2 - The variables assessed and the statistical analysis are mixed together in the Methods. I suggest separating the two aspects in different paragraphs.

3 - Data analysis was carried out in a totally different fashion in the headache clinic and among general practitioners. This is a little bit confusing and leaves doubt regarding the rationale of the study. 

4 - I suggest explaining the rationale for performing the correlation analyses shown in Table 1. In my opinion, those analyses were out of the scope of the present study.

5 - Almost 80% of the headache clinic consultations were general practitioner referrals. Hence, general practitioners might tend to refer patients to a headache center, possibly due to the lack of education about headache management. This is a point to discuss.

6 - Patients referring to a headache center are very different from those treated by general practitioners. Patients treated in headache centers have a high prevalence of chronic headaches, a high number of treatment failures, and a long history of symptoms and medical assessments. Patients treated in general practice might be at their first referral, without previous assessments, and often naïve to treatments. The differences in settings might explain some of the study findings.

7 - The way in which Authors assessed compliance with guidelines is not clear. Were diagnoses and treatment validated by expert physicians?

Round 2

Reviewer 1 Report

The statistical methodology used is now consistent with the study question.

Reviewer 2 Report

I have no further comments.